

# Identification of Gas-phase Pyrolysis Products in a Prescribed Fire: Seminal Detections Using Infrared Spectroscopy for Naphthalene, Methyl Nitrite, Allene, Acrolein and Acetaldehyde**

Nicole K. Scharko[1], Ashley M. Oeck[1], Russell G. Tonkyn[1], Stephen P. Baker[2],
Emily N. Lincoln[2], Joey Chong[3], Bonni M. Corcoran[3], Gloria M. Burke[3], David R. Weise[3],
Tanya L. Myers[1], Catherine A. Banach[1], and Timothy J. Johnson[1*]

[1]Pacific Northwest National Laboratories, Richland, WA, USA
[2]USDA Forest Service, Rocky Mountain Research Station, Missoula, MT, USA
[3]USDA Forest Service, Pacific Southwest Research Station, Riverside, CA, USA

*To whom correspondence should be addressed: Timothy.Johnson@pnnl.gov

## ABSTRACT

Volatile organic compounds (VOCs) are emitted from many sources, including wildland fire; VOCs have received heightened emphasis due to such gases' influential role in the atmosphere, as well as possible health effects. We have used extractive infrared (IR) spectroscopy on recent prescribed burns in longleaf pine stands and herein report seminal detection of five compounds using this technique. The newly reported IR detections include naphthalene, methyl nitrite, allene, acrolein and acetaldehyde. We discuss the approaches used for detection, particularly the software methods needed to fit the analyte and multiple (interfering) spectral components within the selected spectral micro-window(s). We also discuss the method's detection limits and individual species' context in terms of atmospheric chemistry.



## 1. INTRODUCTION

Wildland fire releases significant quantities of trace gases into the environment (Crutzen et al.,
1979; Andreae, 1991; Andreae et al., 2001; Akagi et al., 2011; Yokelson et al., 2013), and such
gases can significantly influence atmospheric chemistry (Crutzen et al., 1990). In some parts of
the world, wildfires are becoming more prevalent as well as increasing in impact (Miller et al.,
2009; Turetsky et al., 2011). In many areas, however, prescribed burning is used as a preventive
tool to reduce hazardous fuel buildups in an effort to reduce or eliminate the risk of such wildfires
(Fernandes et al., 2003). Understanding the products associated with the burning of biomass has
received considerable attention since the emissions can markedly impact the atmosphere. Fourier
transform infrared (FTIR) spectroscopy is one technique that has been extensively used to identify
and quantify gases emitted from burns, generally used in either an open path configuration (Burling
et al., 2010; Akagi et al., 2014; Stockwell et al., 2014; Selimovic et al., 2018) or as an extractive
method (Burling et al., 2011; Akagi et al., 2013; Akagi et al., 2014). Extractive systems typically
use a long-path gas cell coupled to an FTIR instrument so as to increase the sensitivity. Such
approaches have been quite successful; an increasing number of species continue to be identified
and quantified due to the availability of reference gas-phase spectral libraries such as the PNNL
library (Sharpe et al., 2004) or the HITRAN database (Gordon et al., 2017). Such libraries contain
absorption cross-sections that make it possible to obtain quantitative results (i.e. mixing ratios)
without the need for calibration gases. To the best of our knowledge, the actual list of biomass
burning chemical species measured by FTIR has remained limited to ca. 36 compounds (Table 1);
one goal of our research was to expand the list of chemical species to which infrared methods
could be applied. All of the compounds detailed in this study have in fact been previously detected
using other analytical methods (Karl et al., 2007; Yokelson et al., 2009; Akagi et al., 2013; Gilman



et al., 2015; Koss et al., 2018) such as proton-transfer-reaction time-of-flight mass spectrometry
(PTR-ToF) (Koss et al., 2018) or gas chromatography-mass spectrometry (GC-MS) (Gilman et al.,
2015), but have not as yet been identified using FTIR in burning investigations. We wished to
determine if such species' signatures are also found sequestered in the IR spectra associated with
wildland fire, and are thus amenable to IR detection.  A second goal of the present study, whose
biomass burning results are mostly detailed in a separate manuscript, is to better understand
pyrolysis. Every wildland fire consists of two processes: thermal decomposition (pyrolysis) of
solid wildland fuels into gases, tars, and char followed by combustion (oxidation) of the pyrolysis
products resulting in flame gases and particulate matter in the smoke. Description and
measurement (by any means) of the pyrolysis products adjacent to the flames of a wildland fire
has seldom been performed. Non-intrusive measurement of the (pyrolysis) gases in the near-flame
environment is desirable from both a scientific and safety perspective.
The major gas-phase compounds emitted from wildland fires are $H_2O$, $CO_2$, CO and $CH_4$ (Ward
et al., 1991), all of which are easily identified and quantified via FTIR spectroscopy. Lightweight
hydrocarbons, oxygenated hydrocarbons, nitrogen and sulfur species are all minor products that
are also generated during burns (Talbot et al., 1988; Lobert et al., 1991; Yokelson et al., 1996). A
host of more complex gases which can condense to form tar are also produced by pyrolysis of
wildland fuels (Safdari et al., 2018; Amini et al., 2019).  In a gas-phase IR spectrum of such
species, however, the peaks associated with the minor products are often obfuscated by the more
prominent features, such as those from $CO_2$, and can only be recognized in the residual of a
multicomponent simulated fit once the larger features have been removed.  Using data from a
recent field campaign to measure pyrolysis products carried out in a pine forest at Fort Jackson,
South Carolina, we have analyzed some of the IR spectra in more detail to search for the signatures




of compounds not found in Table 1. As a partial guide of species for which to investigate, we
searched for those species detected in previous thermogravimetric-FTIR (TG-FTIR) studies
(Bassilakis et al., 2001; Taghizadeh et al., 2015). TG-FTIR experiments, however, are typically
small-scale and carried out in controlled environments (in contrast to ambient conditions of
prescribed burns or large-scale laboratory burns) and thus represent burns with different oxidative
capacities / combustion efficiencies (Yokelson et al., 1996; Fang et al., 2006; Akagi et al., 2014).
In this study, we have chosen to examine field fire spectra for species that can be detected and
quantified via IR spectroscopy both to add to the list of compounds, but also to improve the
characterization (and ultimately the detection limits) of the other species listed in Table 1.  That is
to say, fire IR spectra are very complex and contain many overlapping peaks; the success of the
spectral analysis depends both on the selected spectral region and proper analysis of all compounds
included in the fit to that domain. The chemometric results become more reliable as the signatures
of all relevant species are included in the fit.
**Table 1.** Compounds previously detected in biomass burning studies using FTIR methods (Yokelson et
al., 1996; Yokelson et al., 1997; Goode et al., 1999; Goode et al., 2000; Christian et al., 2003; Christian et
al., 2004; Karl et al., 2007; Yokelson et al., 2009; Alves et al., 2010; Burling et al., 2010; Burling et al.,
2011; Akagi et al., 2013; Akagi et al., 2014; Stockwell et al., 2014; Gilman et al., 2015; Hatch et al.,
2017; Selimovic et al., 2018).

| Compounds | | | | |
|---|---|---|---|---|
| CO | NO | methanol | phenol | HCOOH |
| $CO_2$ | $NO_2$ | acetic acid | furaldehyde | peroxyacetyl nitrate** |
| $CH_4$ | HONO | $SO_2$ | hydroxyacetone | limonene |
| $C_2H_2$ | $NH_3$ | furan | 1,3-butadiene | carbonyls as glyoxal |
| $C_2H_4$ | HCN | $H_2O$ | acetone | HCHO |
| $C_2H_6$ | HCl | $N_2O$ | isoprene | 2-methylfuran* |
| $C_3H_6$ | $O_3$** | OCS | glycolaldehyde | MVE (methyl vinyl ether) |
| $C_4H_8$ | | | | |

* used in the fit, but not analyzed, ** secondary components found downwind



## 2. EXPERIMENTAL

### 2.1 Site description and sampling device

In early May 2018 seven prescribed fires were conducted in pine forests at U.S. Army Garrison Fort Jackson, adjacent to Columbia, South Carolina, at sites not far from previous smoke emission studies (Akagi et al., 2013; Weise et al., 2015). The forest overstory was primarily longleaf pine (*Pinus palustris* Mill.) and slash pine (*Pinus elliottii* Engelm.), while sparkleberry (*Vaccinium arboreum* Marshall) dominated the understory vegetation. During each burn, pyrolyzed gases emitted at the base of the flames before ignition were collected using an extractive probe and stored in 3-liter Summa canisters. This approach was performed to selectively collect pyrolysis gases prior to the onset of combustion. Details regarding the site description and sampling apparatus will be provided in a separate paper.

### 2.2 FTIR Spectrometer

Gases were analyzed in the laboratory (on the same day or the day following the fire) using an 8-meter multipass (White) cell (Bruker Optics, A136/2-L) mounted in the sample compartment of a Bruker Tensor 37 FTIR. Ten canisters were returned from the field to the laboratory and in turn connected to the gas cell via 3/8" stainless steel tubing. The tubing and gas cell were both heated to 70°C to prevent analyte adhesion to the inner surfaces. The White cell (White, 1942) was equipped with a pressure gauge and temperature probe, both of which were located on the gas outlet port; the thermocouple wire temperature probe extended into the White cell volume in order to more accurately measure the gas temperature. Prior to the start of the series of experiments, it was necessary to calibrate the path length of the variable path gas cell. Measurements of pure isopropyl alcohol (IPA, Sigma-Aldrich, 99.5%) at ten different pressures were collected and a Beer's Law plot was created to determine the length. The IR region from 3515 to 3290 cm$^{-1}$ was




integrated (Figure 1a), and the corresponding areas plotted as a function of the IPA pressure
(converted to ppm at 760 Torr)  multiplied by the PNNL reference library (Sharpe et al., 2004)
integration area for a 1 ppm-meter IPA burden (Figure 1b). The slope is equal to the path length,
which was determined to be 8.10 m.

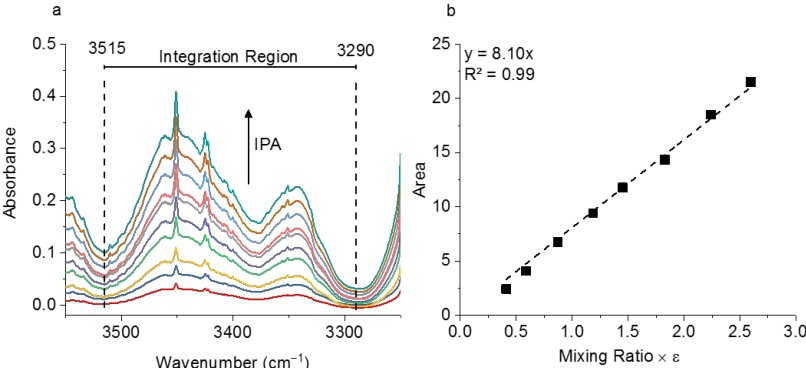


**Figure 1.** a) Multiple burden spectra of dry IPA for 10 measurements at varying pressures. The dashed
lines represent the integration limits used for spectral integration. b) Calibration plot with regression line
for IPA measurements. The slope of the line is the path length in meters.

The White cell contained the analyte smoke for the sample spectrum measurement, but was filled
with ultra-high purity nitrogen gas for the reference spectrum measurement (Johnson et al., 2013).
The FTIR interferometer, detector and sample compartments were purged with dry air from a dry-
air generator. The Tensor 37 was equipped with a globar source, a KBr beamsplitter and a
broadband liquid nitrogen cooled mercury cadmium telluride (MCT) detector, providing spectral
coverage from 7,500 to 500 cm$^{-1}$.  The spectral resolution was 0.6 cm$^{-1}$ and a 2 mm Jacquinot
aperture was used. The acquisition mode was set to double-sided, forward-backward. For the
Fourier transform, the data were apodized with a Blackman-Harris 3-Term function using a zerofill
factor of 4, and phase corrected via the Mertz (Mertz, 1967) method.



**2.3 Quantitative Spectral Analysis**
The program used for quantitative spectral analysis was MALT5 (Griffith, 2016), which uses both
broadband reference spectra from PNNL (Sharpe et al., 2004; Johnson et al., 2006; Johnson et al.,
2009; Profeta et al., 2011; Lindenmaier et al., 2017) and absorption line intensities from HITRAN
(Gordon et al., 2017) (in units of $cm^{-1}/(\text{molec} \times cm^{-2})$) to iteratively fit a simulated spectrum to
the measured spectrum by optimizing the fit so as to minimize the mean-squared residual, i.e. the
difference between the measured and simulated spectra. Parameters such as path length, resolution,
apodization, temperature, pressure, spectral domain, target compounds / overlapping compounds
are all used as inputs to the spectral fit. During the course of this study, MALT5 was used to
identify five gas-phase species emitted during the burns and quantify the gas mixing ratios via IR
spectroscopy for the first time. Part of the confirmation strategy is to process the experimental
spectra both with and without the target compound present in the fit and visually inspect the
corresponding residuals. Table 2 summarizes the IR-active vibrational mode used for each species
in the spectral fit (typically the species' strongest band in the longwave infrared window), along
with the spectral domain and a list of species with overlapping bands in that domain.








**Table 2.** Gas-phase species identified via FTIR, vibrational assignments (Lord et al., 1952; Hollenstein et al., 1971; Ghosh et al., 1981; Hamada et al., 1985; Es-Sebbar et al., 2014; Chakraborty et al., 2016), and spectral domains used for spectral fit and quantitation.

| Target compound | Vibrational bands used for analysis | Spectral region ($cm^{-1}$) | Other species fit in the same region |
|---|---|---|---|
| Naphthalene | $\nu_{46}$ at 782.3 $cm^{-1}$ | 800–760 | $C_2H_2$, $CO_2$, HCN and $H_2O$ |
| Methyl nitrite | $\nu_8$ at 841.1 (*cis*) and 812.3 (*trans*) $cm^{-1}$ | 865–775 | $C_2H_2$, $CO_2$, HCN, naphthalene, $C_2H_4$, allene, and $H_2O$ |
| Allene | $\nu_{10}$ at 845.3 $cm^{-1}$ | 865–775 | $C_2H_2$, $CO_2$, HCN, naphthalene, $C_2H_4$, methyl nitrite, and $H_2O$ |
| Acrolein | $\nu_{10}$ at 1157.7 $cm^{-1}$ | 1200–1100 | Acetic acid ($CH_3COOH$), furfural ($C_4H_3OCHO$), acetaldehyde , HCOOH, $CH_4$, $C_2H_4$, and $H_2O$ |
| Acetaldehyde | $\nu_3$ at 2716.2 $cm^{-1}$ | 2800–2650 | $CH_4$, HCHO, $C_2H_2$, acrolein, and $H_2O$ |

## 2.4 Spectral Resolution

As mentioned in section 2.2, the spectral resolution was set to 0.6 $cm^{-1}$, which is the highest resolution obtainable with this instrument. There are many benefits, but also a few disadvantages to using higher resolution (Herget et al., 1979). Most importantly, the higher resolution allows one to resolve the narrow bands of key analytes and discriminate them from lines or bands of interferents. For example, in the present study the 782 $cm^{-1}$ Q-branch of naphthalene was distinguished from the adjacent absorption lines of $C_2H_2$ [Naphthalene's IR bands and results are discussed in greater depth in Section 3.1]. If a lower resolution were used, then the deconvolution of naphthalene from $C_2H_2$ would have been compromised, perhaps unfeasible. To demonstrate, one of the experimental measurements collected at a resolution of 0.6 $cm^{-1}$ was deresolved to 1, 2, and 4 $cm^{-1}$ using a Gaussian profile as seen in Figure 2. Those spectra were processed by MALT5 to check for the presence of naphthalene. Figure 2 displays the measured spectra and the scaled reference spectra for $C_2H_2$ and naphthalene, and the corresponding residuals with and without naphthalene included in the fit for the a) original spectrum collected at 0.6 $cm^{-1}$ and the





deresolved spectra at b) 1 cm$^{-1}$, c) 2 cm$^{-1}$, and d) 4 cm$^{-1}$. With the reference spectra for the original
0.6 cm$^{-1}$ measurement and the 1 cm$^{-1}$ deresolved spectrum (Figure 2a and b), the absorption lines
for C$_2$H$_2$ and naphthalene overlap, but the 782 cm$^{-1}$ feature from naphthalene is still slightly visible
in the original spectra. The naphthalene peak appears clearly in the residuals when it is not included
in the fitting process, but is removed from the residual when naphthalene is included in the fit
(discussed further below). As the resolution is reduced (Figures 2c and 2d), however, the features
broaden and the distinction of the naphthalene peak from C$_2$H$_2$ and other minor components (i.e.
CO$_2$, HCN, H$_2$O, spectra not shown) is compromised. The specificity between the compounds is
lost and the confidence in the identification/quantification of the target species, particularly for the
weaker absorbers, diminishes as the resolution decreases. The well-known benefits of using a
lower resolution are that spectra can be acquired more quickly with an improved signal-to-noise
ratio. For the present measurements, 0.6 cm$^{-1}$ was deemed an appropriate resolution.





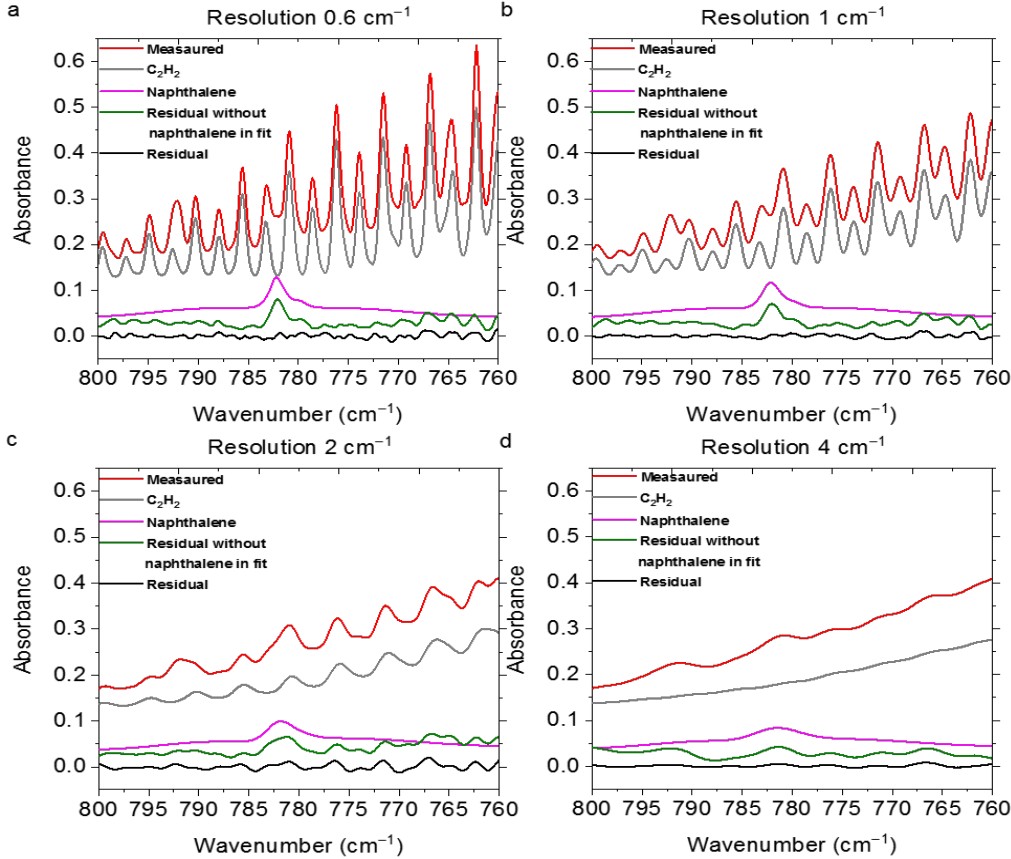

**Figure 2.** Measured and scaled reference spectra for $C_2H_2$ and naphthalene, and corresponding residuals with and without naphthalene included in the fit for the a) original spectrum collected at 0.6 cm$^{-1}$ and the deresolved spectra at b) 1 cm$^{-1}$, c) 2 cm$^{-1}$, and d) 4 cm$^{-1}$. The reference spectra for $CO_2$, HCN and $H_2O$ are not shown (HCN was not included in fit when the resolution was 4 cm$^{-1}$; for resolutions 1, 2 and 4 cm$^{-1}$, $H_2O$ was not included in the fit when naphthalene was removed from the fit). Spectra are offset for clarity.

## 2.5 Detection Limits and Signal-to-noise Ratio

The detection limit values presented in this paper are not minimal signal-to-noise limits in the sense of a minimal spectral signal against a background of purely stochastic noise sources. In such cases, the noise sources are typically of comparable or higher frequencies than the signal (Johnson et al., 1991). Rather, the current limits represent the average detection limits for a spectral residual derived from a convoluted spectrum arising from a gas mixture of differing and fluctuating chemical composition. The residuals are due to the least-squares fit of (fluctuations in) the many



complex features arising from numerous chemicals. That is to say, the residual is not due to just
random instrumental noise, but instead, due to spectral features that can arise in the spectra, e.g.
imperfectly subtracted features from strong absorbers or unidentified absorbers.  For that reason,
we report signal-to-residual, not signal-to-noise detection limits. The detection limits for each
compound in this study were thus derived using a value of three times the root-mean-square (RMS)
value of the residual calculated over the corresponding frequency range (e.g. 800–760 $cm^{-1}$ was
used for naphthalene). The peak-to-peak noise is more sensitive to fluctuations in the fit with levels
typically 4 to 5× the RMS noise (Griffith et al., 2006).  For the present data, however, the peak-to-
peak values ranged from 5 to 10× the RMS noise, thus suggesting the peak-peak values tend to
overstate the tractable noise level, i.e. understate the detection limit. The reported detection limits
are thus presumably higher than what would be estimated with an FTIR in clean air conditions (i.e.
only the analyte and dry air). Based on experience, the limits are typically far higher than what can
be obtained with IR laser sensors where the intrinsically narrow laser linewidths allow for the
probing of individual rotational-vibrational lines without drawing in overlapping spectral lines to
a congested spectral fit (Taubman et al., 2004; Wagner et al., 2011; Phillips et al., 2014). While
typically far more sensitive, such laser measurements can only analyze for one or a few species at
a time, as opposed to the 30+ species seen by the broadband FTIR measurements.
**3. RESULTS AND DISCUSSION**
When modeling the burning process (Byram, 1959), complete combustion of 1 kg dry wood
produces 1.82 kg $CO_2$ and 0.32 kg $H_2O$ for a total mass of products of 2.14 kg.  Incomplete
combustion will yield additional products and less $CO_2$ and $H_2O$ while combustion of wet fuels
(Byram, 1959) increases the amount of $H_2O$ released.  For infrared analysis of such smoke, much
of the challenge arises due not only to the large mole fractions of $H_2O$ and $CO_2$, but the fact that



both $H_2O$ vapor and $CO_2$ have strong features in the mid-IR that can clutter the spectrum rendering
certain spectral regions unusable. For burning and other atmospheric studies, ideal compounds for
detection via IR spectroscopy will thus have strong absorption coefficients that do not overlap with
the fundamental bands of $H_2O$ or $CO_2$, i.e. are in a spectral window or microwindow (Griffith,
1996; Esler et al., 2000; Smith et al., 2011) free of strong interferences. Here, we consider five
such compounds emitted during this prescribed burn, but which had heretofore not been reported
as being detected by FTIR. Individual compounds are discussed in turn regarding their formation
mechanism(s), detectable IR features and spectral confirmation for this study, along with their
potential fates and atmospheric impacts. Lastly, the results are briefly compared with literature
values using emission ratios (mixing ratios of analyte to excess CO).

**3.1 Naphthalene**

Naphthalene ($C_{10}H_8$) is a polycyclic aromatic hydrocarbon (PAH) that is emitted from certain
chemical industries as well as from the combustion of gasoline and oil (Jia et al., 2010). It is a
condensable hydrocarbon also generated by biomass pyrolysis (Liu et al., 2017). There are a
number of pyrolysis formation routes (Fairburn et al., 1990; Williams et al., 1999; Richter et al.,
2000; Lu et al., 2004; Liu et al., 2017). One proposed mechanism is the generation of single ring
aromatic compounds such as benzene, toluene and styrene via Diels-Alder reaction of alkenes; the
single ring aromatic compound then combines with alkenes to form double-ring PAHs, such as
naphthalene (Fairburn et al., 1990). Naphthalene may even undergo subsequent reactions to form
still larger polyaromatics (Fairburn et al., 1990; Richter et al., 2000). Naphthalene has been
detected (via GC-MS) in tars that were condensed from gas-phase pyrolysis products of both live
and dead southeastern fuels, such as live oak (*Quercus virginiana*) and swamp bay (*Persea*
*palustris*) (Safdari et al., 2018). It has been also detected (Hosseini et al., 2014; Aurell et al., 2017;



Koss et al., 2018) in the gas-phase in laboratory burning experiments. The detection of gas-phase
naphthalene from wildland fire emissions is thus not surprising, but this is the first report of its
identification via IR spectroscopy. The best spectral feature for identification and quantification is
the $\nu_{46}$ IR mode near 782.3 cm$^{-1}$, which corresponds to the H–C–C out-of-plane bend (Chakraborty
et al., 2016). There are other bands at 3067.7 and 3058.0 cm$^{-1}$ previously assigned to $\nu_{29}$ and $\nu_{17}$,
respectively (Chakraborty et al., 2016). Both of these modes have smaller absorption coefficients
as compared to $\nu_{46}$, however, and are located in the C–H stretching region, which is common to
nearly all hydrocarbons and thus provides less specificity.
Figure 3 shows a prescribed burn spectrum in the region from 800 to 760 cm$^{-1}$. The primary
spectral signatures in this plot are those of the R-branch rotational-vibrational lines associated with
the $\nu_5$ fundamental (Kabbadj et al., 1991) of $C_2H_2$, but there are also absorptions due to $CO_2$, HCN,
$H_2O$ (individual spectral contributions not shown) and naphthalene. When all of the spectral
components except for naphthalene are included in the fitting process, the residual (green trace)
displays a prominent feature at 782.3 cm$^{-1}$, which we ascribe to naphthalene. When naphthalene
is included in the fit, the feature in question is removed as seen in the black trace of Figure 3.





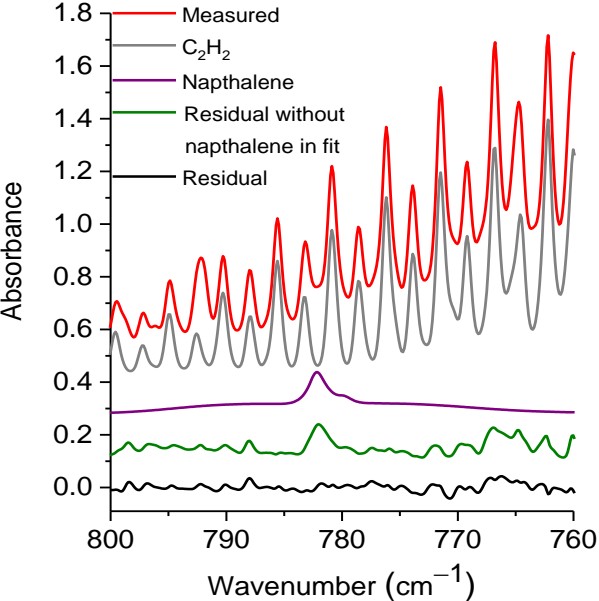


**Figure 3.** Measured spectrum, scaled reference spectra for $C_2H_2$ and naphthalene, and residuals
with and without naphthalene included in the fit. For clarity, the spectral contributions for $CO_2$,
HCN, and $H_2O$ are not shown. All spectra are at 0.6 cm$^{-1}$ resolution and have been offset. The
calculated mixing ratio of naphthalene in this measured spectrum is $16.4 \pm 0.6$ ppm (values obtained
from MALT5 software, and error represents standard error).



Table 3 presents the range of measured mixing ratios for naphthalene along with averaged
detection limits for the 10 measurements collected during the prescribed burns as well as for the
other four reported compounds. In the measurements, naphthalene's mixing ratios ranged from 1.4
to 19.9 ppm, and the averaged RMS-derived detection limit was $1.6 \pm 0.5$ ppm; different detection
limits were observed for each spectrum. One of the measurements had a mixing ratio of 2.9 ppm,
yet its RMS-derived detection limit was 3.7 ppm, and is thus below the estimated detection limit
(bdl).






**Table 3.** Calculated mixing ratios for ten canister FTIR measurements along with average estimated residual detection limits for the target compounds derived using 3 times the root-mean-square of the residual. Error bars represent the standard deviation (1σ) of the mean.

| Target compound | Calculated mixing ratios (ppm) | | | Averaged detection limit (ppm) using root-mean-square (RMS) value of the residual |
|---|---|---|---|---|
| | Min | Max | Average | |
| Naphthalene* | 1.4 | 19.9 | 8.5 ± 2.1 | 1.9 ± 0.5 |
| Methyl nitrite* | 2.3 | 21.0 | 8.7 ± 2.4 | 2.2 ± 0.4 |
| Allene | 2.2 | 37.8 | 13.1 ± 3.6 | 3.0 ± 0.6 |
| Acrolein | 14.7 | 125.7 | 43 ± 12 | 6.1 ± 1.5 |
| Acetaldehyde | 34.5 | 264.8 | 103 ± 27 | 11.7 ± 3.2 |

*One measurement was below the detection limit.

Naphthalene emitted from prescribed burns is thus clearly detectable using IR spectroscopy. The

U.S. Environmental Protection Agency considers naphthalene a potential human carcinogen and a

hazardous air pollutant (U.S. EPA). Once released, naphthalene may cycle in the atmosphere or

accumulate in aquatic and terrestrial systems via wet/dry deposition or air-water gas exchange

(Park et al., 2001). Gas-phase naphthalene's primary atmospheric loss mechanism is its reaction

with the hydroxyl radical (OH) to form hydroxy-PAHs or nitro-PAHs in the presences of nitrogen

oxides (Vione et al., 2004). The estimated atmospheric lifetime of naphthalene for reaction with

OH is 6.8 hours (based on a 12-hour daytime OH level of $1.9 \times 10^6$ molecules cm$^{-3}$) (Arey, 1998).

**3.2 Methyl Nitrite**

A second compound detected for the first time in the wildland fire IR spectra was methyl nitrite

($CH_3ON=O$). Methyl nitrite has previously been observed in aged cigarette smoke (Schmeltz et

al., 1977), and also the exhaust of engines fueled by methanol–diesel blends (Jonsson et al., 1982).

It has also been observed as a minor product for the thermal decomposition of both nitrate esters

(Boschan et al., 1955) and isopropyl nitrate at low temperatures and pressures (Griffiths et al.,

1975). Methyl nitrite has also been detected in wildland fire emissions by GC-MS (Gilman et al.,





2015). Moreover, it has been observed that some nitrogen-containing organic compounds such as
acetonitrile ($CH_3CN$) and acrylonitrile ($CH_2=CHCN$) emitted from burns were directly correlated
to the fuel nitrogen content. However, methyl nitrite [and another oxygenated nitrogen organic
compound, isocyanic acid (HNCO)] did not show any significant dependency on fuel N-content
(Coggon et al., 2016). It has been suggested that methyl nitrite is only a minor direct product of
combustion (Finlayson-Pitts et al., 1992),  but instead is generated *in situ* by the secondary reaction
of methanol ($CH_3OH$) with nitrogen dioxide ($NO_2$).
We also note that methyl nitrite is an oxidizing agent and is used as a rocket propellant. It is thus
plausible that the methyl nitrite detected in the present study was not a product of the fire, but
emanated from munitions used in training at Ft. Jackson. However, while the records of the
munitions used were not complete, a survey of these records did not indicate the use of methyl
nitrite in any munitions at the Ft. Jackson plots where the present burn samples were collected.
With regards to the IR spectra, methyl nitrite exists in equilibrium as a mixture of two conformers-
*cis* and *trans*; at room temperature (25°C) it is estimated as 58% *cis* and 42% *trans* (Bodenbinder
et al., 1994). We were able to use the same band associated with both conformers, namely the $\nu_8$
band, which is at 841.1 $cm^{-1}$ for the *cis* conformer and at 812.4 $cm^{-1}$ for the *trans* conformer
(Ghosh et al., 1981). The $\nu_8$ mode is associated with the N–O stretch and is very strong for both
conformers (Ghosh et al., 1981). We note that methyl nitrite also has very strong bands at 627.8
$cm^{-1}$ (*cis*) for $\nu_9$ ONO bending, as well as at 1620.1 $cm^{-1}$ (*cis*) and 1677.4 $cm^{-1}$ (*trans*) due to the
$\nu_3$ N=O stretch (Ghosh et al., 1981). These bands, however, are of lesser utility for IR detection:
The $\nu_9$ peak is masked by $CO_2$ bending mode lines, and the $\nu_3$ peak is obfuscated by the $H_2O$
bending mode lines.



The spectral region used for evaluation was 865–775 cm$^{-1}$, which contains the $\nu_8$ band for both the
*cis* and *trans* conformers (Ghosh et al., 1981). Figure 4 shows the experimental spectrum from
the prescribed burn, along with the scaled reference spectra for the two major compounds used in
the fit: $C_2H_2$ and methyl nitrite. While important, other minor compounds, such as $CO_2$, HCN,
naphthalene, $C_2H_4$, allene, and $H_2O$, were also included in the analysis, but their spectral
contributions are not plotted. Additionally, Figure 4 displays the residuals both when methyl nitrite
was included in the fitting process and when it was excluded. Upon inspection of the residual
spectrum where it was excluded (green trace), it is clear that both the *cis* and *trans* features from
$\nu_8$ are present and this confirms methyl nitrite in the pyrolysis smoke

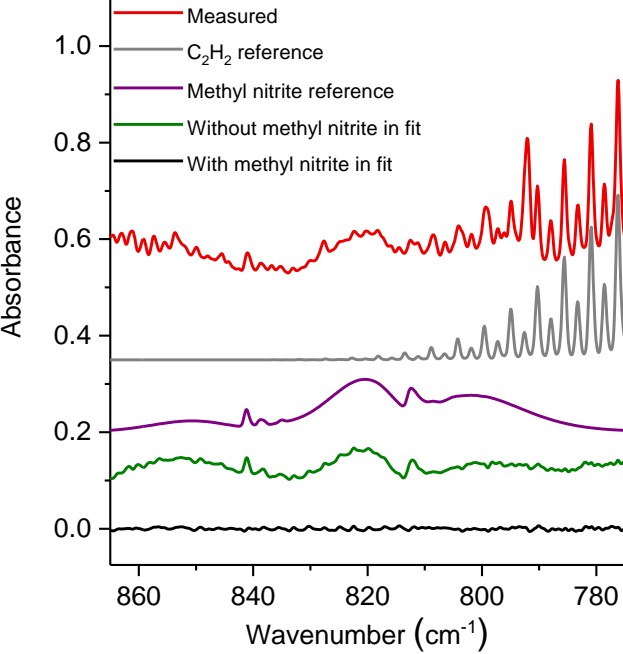

**Figure 4.** Measured experimental spectrum and the individual spectral contributions for the major
components ($C_2H_2$ and methyl nitrite) and residuals with and without methyl nitrite included in the fit. For
clarity, the spectral contributions for $CO_2$, HCN, naphthalene, $C_2H_4$, allene, and $H_2O$ are not shown. All
spectra are at 0.6 cm$^{-1}$ resolution and have been offset for clarity. The calculated mixing ratio of methyl
nitrite in this measured spectrum is 21.0 ± 0.1 ppm (values obtained from MALT5 software, and error
represents standard error).



The mixing ratio and RMS-derived detection limit for methyl nitrite for the displayed experimental
spectrum in Figure 4 are 21.0 ppm and 1.4 ppm, respectively. The range for the mixing ratios and
the averaged detection limits for methyl nitrite are summarized in Table 3. Methyl nitrite was
detected with confidence in 9 of the 10 measurements; only one of the measurements was below
the RMS-derived detection limit.
We report the detection via IR spectroscopy of methyl nitrite in wildland fire emissions not only
because it is novel, but also because of its influential role in atmospheric chemistry: Methyl nitrite
is a photochemical source of OH. In the atmosphere it undergoes photolysis to form the methoxy
radical ($CH_3O$) and nitric oxide (NO) with a quantum yield near unity (Cox et al., 1980). At solar
noon, the photolytic lifetime is only 10–15 min (Seinfeld et al., 2012). The photogenerated
methoxy radical then undergoes subsequent reactions leading to the formation of OH. In turn, both
OH and NO contribute to the production of ozone (Finlayson-Pitts et al., 1999).
**3.3 Allene**
Allene (1,2-propadiene, $CH_2=C=CH_2$) is of high symmetry ($D_{2d}$) and has the two methylene
groups with their H–C–H planes at right angles to each other (Lord et al., 1952). The compound
has previously been detected in biomass burning grab samples using GC (Akagi et al., 2013).
Allene is a proposed precursor in the burning process that contributes to the formation of both
aromatic compounds and soot (Frenklach et al., 1983; Frenklach et al., 1988). Lifshitz et al. have
observed (at temperatures ranging from 757–847°C) that the structural isomerization of allene and
propyne ($CH_2=C=CH_2 \leftrightarrow CH_3–C≡CH$) will take place via a unimolecular reaction faster than the
decomposition reaction (Lifshitz et al., 1975). Additionally, these same authors investigated the
pyrolysis of allene and propyne and observed that $C_2H_4$ was generated from allene while $CH_4$ and
$C_2H_2$ were mainly formed from propyne (Lifshitz et al., 1976). Unfortunately, the strongest IR



band for propyne (near 634 cm$^{-1}$) is obscured by $CO_2$ bending mode lines. Due to the interferences
we cannot with confidence identify propyne in the measurements; we can, however, detect allene.
In the mid-IR, allene has several strong rotational-vibrational lines near 845 cm$^{-1}$ associated with
the sub-bands of the perpendicular band $\nu_{10}$, which is due to $CH_2$ rocking (Lord et al., 1952).
Additionally, allene has a moderately strong band at 1958.6 cm$^{-1}$ due to the $\nu_6$ C–C stretching
(Lord et al., 1952). However, the $\nu_6$ band is not useful for detection due to interference from the
$H_2O$ bending mode lines.

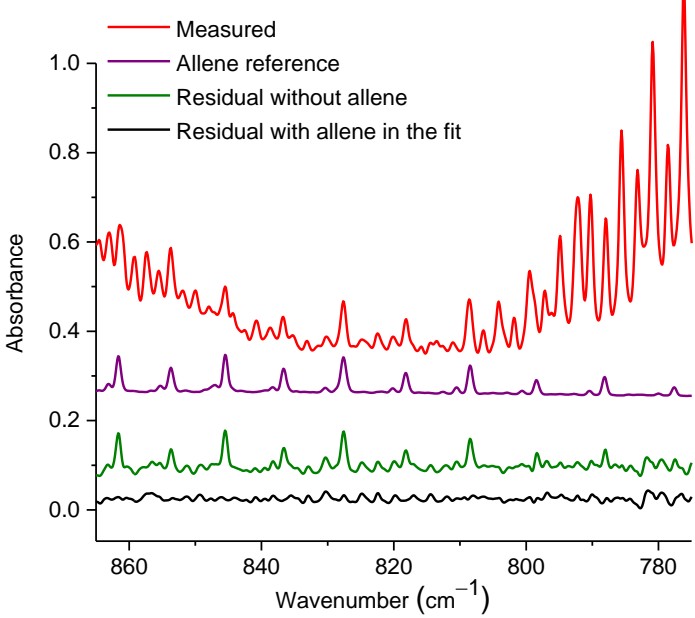

**Figure 5.** Measured absorbance spectrum and residual with and without allene included in the fit, along
with the scaled reference spectrum for allene. For clarity, the spectral contributions for $C_2H_2$, $CO_2$, HCN,
naphthalene, $C_2H_4$, methyl nitrite, and $H_2O$ are not shown. All spectra are at 0.6 cm$^{-1}$ resolution and have
been offset for clarity. The calculated mixing ratio of allene in this measured spectrum is 37.8 ± 0.6 ppm
(values obtained from MALT5 software, and error represents standard error).

Figure 5 shows the measured absorbance spectrum, scaled allene reference spectrum and the
associated residual with and without allene included in the fit. The absorption lines associated
with allene are clearly seen in the resulting spectrum when allene is not included in the fit (green



trace), thus confirming that allene is one of the primary components contributing to the features in
this spectral domain. For the experimental spectrum displayed in Figure 5, the calculated mixing
ratio for allene is 37.8 ppm and the RMS-derived detection limit is 5.4 ppm.

Unlike naphthalene and methyl nitrite, allene is not considered a hazardous air pollutant nor is it a
photochemical source of OH. Major loss processes for alkenes include reactions with OH, $NO_3$
radical and $O_3$ (Atkinson et al., 2003). Specifically for allene, the lifetime (calculated from rate
constants from Atkinson et al., 2003, and based on a 12-hour daytime OH level of $1.9 \times 10^6$
molecules $cm^{-3}$ and 24-hour $O_3$ average of $7 \times 10^{11}$ molecules $cm^{-3}$) with respect to OH and $O_3$
reactions are 1.2 and 89.4 days, respectively. The reaction between OH and allene involves the
initial addition of OH to one of the C=C bonds generating a hydroxyalkyl radical, which then may
undergo subsequent reactions (i.e. reaction with $O_2$ forming hydroxyalkyl peroxy radical)
contributing to the propagation of radicals in the atmosphere (Atkinson et al., 2003; Daranlot et
al., 2012).

**3.4 Acrolein and Acetaldehyde**
The two aldehydes, acrolein ($CH_2=CHCHO$) and acetaldehyde ($CH_3CHO$), have also been
identified in the burning IR spectra. It has been proposed that both acrolein and acetaldehyde are
formed from the pyrolysis of cellulose (a major constituent of biomass) via the intermediate
glycerol, which is a moiety in the structure of levoglucosan, a known pyrolysis product of cellulose
(Stein et al., 1983). Stein et al. observed that acrolein, acetaldehyde and CO were the initial
decomposition products for the pyrolysis of glycerol (Stein et al., 1983). Both of these compounds





have been detected in previous wildland fires studies via methods such as GC (Akagi et al., 2013)
or PTR-ToF (Brilli et al., 2014; Koss et al., 2018), but have not yet been identified via IR.
Acrolein, the simplest unsaturated carbonyl, exists in two forms,  *s-cis* and *s-trans*,  with *s-trans*
being the more stable, and consequently the more abundant conformer (Wagner et al., 1957). It
has been estimated that the fractions of *s-cis* and *s-trans* are about 4 and 96% at 20°C, and 7 and
93% at 100°C, respectively (Alves et al., 1971). The largest IR feature for acrolein is the $\nu_5$ C=O
stretch (Hamada et al., 1985) at 1724.1 cm$^{-1}$, but this band is heavily overlapped by water lines.
There is also the $\nu_{16}$ band (Hamada et al., 1985) at 958.8 cm$^{-1}$, but this feature  overlaps with
multiple other strongly absorbing compounds, such as $C_2H_4$. We have therefore focused acrolein's
analysis using the $\nu_{10}$ band (C–C stretch) (Hamada et al., 1985) at 1157.7 cm$^{-1}$.

Figure 6 displays the very congested biomass burning spectrum with individual contributions for
several species included in the fit [contributions for furfural ($C_4H_3OCHO$), acetaldehyde, $CH_4$, and
$C_2H_4$ are included, but not plotted] as well as the residual with and without acrolein included in
the fitting process. When acrolein is not included in the fit, features (both near 1168 and at 1157.7
cm$^{-1}$) that resemble acrolein are observed in the residual spectrum as seen in the green trace in
Figure 6. When acrolein is included in the fit, the features in question are removed. For acrolein,
no mixing ratios were observed below the RMS-derived detection limits.



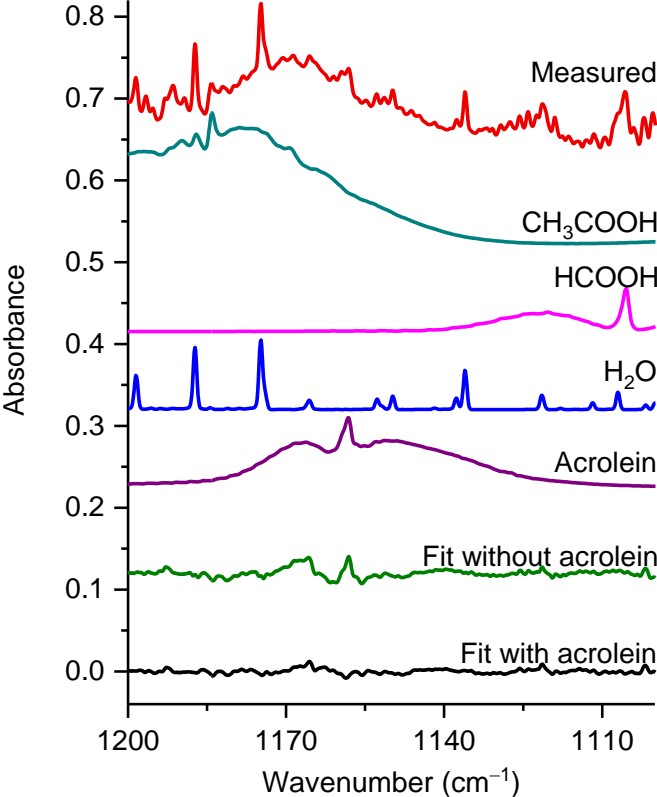

**Figure 6.** Measured spectrum and the individual spectral contributions for the major components and
associated residual with and without acrolein included in the fit. For clarity, the spectral contributions for
furfural ($C_4H_3OCHO$), acetaldehyde, $CH_4$, and $C_2H_4$ are not shown. All spectra are at 0.6 cm$^{-1}$
resolution and have been offset for clarity. The calculated mixing ratio of acrolein in this measured spectrum
is 99.9 ± 3.0 ppm (values obtained from MALT5 software, and error represents standard error).
Similar to acrolein, acetaldehyde has its strongest IR feature due to the C=O stretch (Hollenstein
et al., 1971), with $\nu_4$ found at 1746.1 cm$^{-1}$. Again, due to the presence of water lines in the
spectrum, this features is not practical for detection. The aldehyde $\nu_3$ C–H stretching band
(Hollenstein et al., 1971) at 2716.2 cm$^{-1}$ was instead used for analysis. Figure 7 shows the
measured and fitted spectra as well as the spectral contributions of the major individual
components used to calculate the fitted spectrum and the corresponding residual. Other minor




components, such as acrolein, $C_2H_2$ and $H_2O$, were also included in the fit, but their reference
spectra are not displayed in Figure 7. The spectral profile of acetaldehyde with its P and R branches
of $\nu_3$ is easily discernable even before deconvolution of the measured spectrum. Similar to
acrolein, all of the mixing ratios for acetaldehyde were above the RMS-derived detection limit.

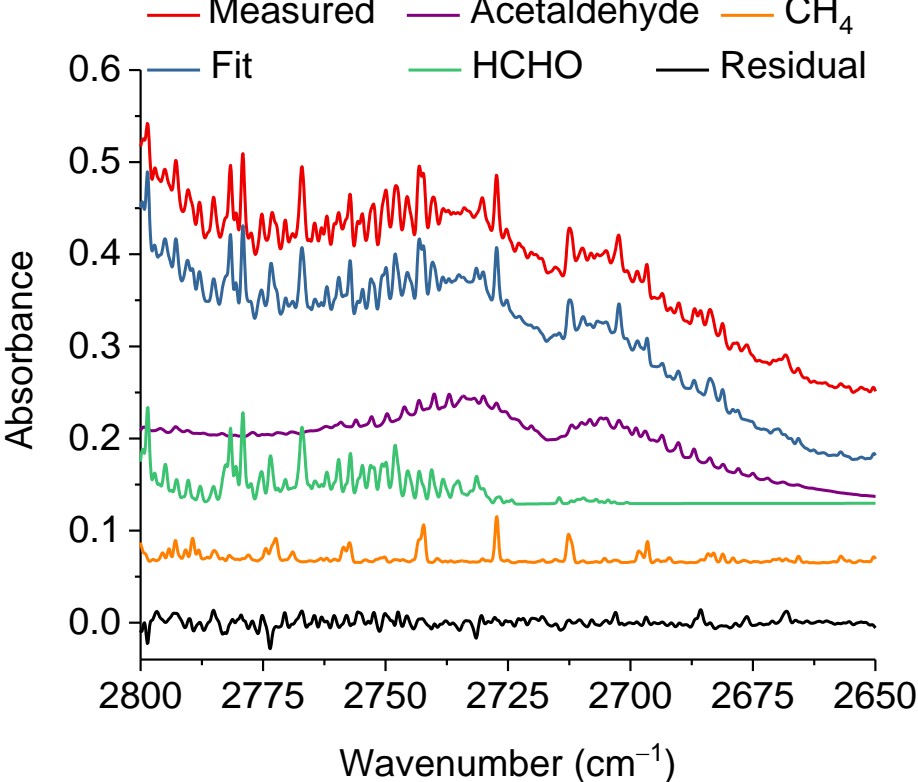

**Figure 7.** Measured and fitted spectra as well as the individual components (for clarity, the spectral
contributions for acrolein, $C_2H_2$, $H_2O$ are not shown) and associated residual in the spectral region 2800-
2650 cm$^{-1}$. All spectra are at 0.6 cm$^{-1}$ resolution and have been offset for clarity. The calculated mixing
ratio of acetaldehyde in this measured spectrum is 252.8 ± 5.5 ppm (values obtained from MALT5 software,
and error represents standard error).
Similar to naphthalene, the U.S. EPA considers both acrolein and acetaldehyde to be hazardous
air pollutants (U.S. EPA). Acrolein is toxic to humans, and when inhaled may cause upper
respiratory irritation. Acetaldehyde will irritate the eyes, skin and the respiratory tract and is




considered a potential human carcinogen (U.S. EPA). Once released into atmosphere, such
aldehydes can either react with $O_3$ or OH, or undergo photolysis (Seinfeld et al., 2012). For
acrolein, OH reaction is the major loss process with a lifetime of 2.4 hours based on a 12-hour
daytime OH level of $1.9 \times 10^6$ molecules cm$^{-3}$, (Gierczak et al., 1997) forming products such as
CO, $CO_2$, HCHO, glycolaldehyde (Johnson et al., 2013) and acryloylperoxynitrate (APAN)
(Orlando et al., 2002). Similarly, acetaldehyde's lifetime is dominated by OH loss, and that
reaction generates HCHO and CO as well as peroxyacetylnitrate (PAN) (D'Anna et al., 2003).
Acetaldehyde's estimated tropospheric lifetimes with respect to OH reaction and photolysis are 10
hours (Atkinson et al., 2003) and 5 days (Seinfeld et al., 2012), respectively.

**3.5 Comparison to Other Measurements**

Preliminary emission ratios (relative to CO) for the reported compounds are compared to those
reported in previous wildland burning investigations: Table 4 displays the average emission ratios
and the standard deviations (1σ) for this study as well as emission ratios reported by Koss et al.
(2018), Ferek et al. (1998), Brilli et al. (2014), and Gilman et al. (2015). As shown in the table,
there is significant variation between the studies due to multiple factors such as different fuel types,
analytical methods, sampling approaches and experimental conditions. For example, the study by
Ferek et al. (1998) focused on the collection of airborne samples, while Brilli et al. (2014)
measured gases under nocturnal conditions using a ground-based system. Inspection of the table
shows that the measured emission ratio values are not unprecedented, but are within range of
previous measurements. Because they have the same molar mass, the mass spectrometric
techniques in some cases cannot distinguish allene from propyne.





**Table 4.** Emission ratios relative to CO and standard deviations (1σ) for the present study and for three other previously published biomass burning studies.

| Target compounds | Present average emission ratios to CO (ppb/ppm) | Koss et al. (2018) fire-integrated emission ratio to CO (ppb/ppm) | Ferek et al. (1998) emission ratio to CO (ppb/ppm) | Brilli et al. (2014) emission ratios to CO (ppb/ppm) | Gilman et al. (2015) discrete emission ratios to CO (ppb/ppm) | | |
|---|---|---|---|---|---|---|---|
| | | | | | South-western fuels | South-eastern fuels | Northern fuels |
| Method | FTIR | PTR-ToF-MS | GC-FID* | PTR-ToF-MS | GC-MS | GC-MS | GC-MS |
| Naphthalene | 0.79 (0.47) | 0.20 (0.16) | n/a | n/a | 0.0070 (0.0048) | 0.0040 (0.0050) | 0.022 (0.012) |
| Methyl nitrite | 0.94 (0.85) | n/a | n/a | n/a | 0.9 (1.1) | 0.52 (0.51) | 0.76 (0.90) |
| Acrolein | 4.0 (1.8) | 5.4 (3.0) | n/a | 3.14 (0.12) | 0.82 (0.68) | 1.31 (0.88) | 3.5 (1.7) |
| Acetaldehyde | 9.4 (3.6) | 7.4 (5.2) | n/a | 37.3 (1.4) | 1.6 (1.2) | 2.8 (1.8) | 5.5 (3.6) |
| Allene (Propadiene)** | 1.05 (0.24) | n/a | 0.1 (0.1) | 8.73 (0.28) | n/a | n/a | n/a |

*GC-FID is gas chromatography with flame ionization detector
**Brilli et al. (2014) use both 1-propyne and propadiene to represent $C_3H_4$. Gilman et al. (2015) report emission ratios for propyne, but not allene.

## 4. SUMMARY

Gas-phase compounds with appreciable band intensities and appreciable concentrations can be both identified and quantified using IR spectroscopy. We have used such spectral information for seminal IR detection of five compounds generated during prescribed forest fire burns. Deriving the mixing ratios from the congested spectra obtained from wildland smoke samples is more challenging due to the multiple overlapping spectral features: Sophisticated software and analysis are required in carefully selected spectral windows. We have reported seminal IR detection of five molecules that had previously not been observed by FTIR in ambient measurements of wildland emissions. Most of the compounds (excluding acetaldehyde), had their primary features become apparent only after the larger spectral features had been fitted and subtracted.



## ACKNOWLEDGMENT

We gratefully acknowledge support from the Department of Defense's Strategic Environmental

Research and Development Program (SERDP), Project RC-2640 and gratefully thank our sponsor

for their support. PNNL is operated for the U.S. Department of Energy by the Battelle Memorial

Institute under contract DE-AC06-76RLO 1830. We thank Professor Valerie Young of Ohio

University for loan of the canisters. 1830. We thank Prof. David W. T. Griffith for his valuable

guidance and direction using the MALT5 program for spectral analysis. We kindly thank John

Maitland and colleagues at Fort Jackson for conducting the burns and hosting the scientific

mission, and Olivia Williams for assistance with the MALT calculations. Lastly, we are most

grateful to Professor Michael L. Myrick and his colleagues at the University of South Carolina for

allowing us to use their laboratories and for their helpful assistance during the campaign.

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
