# Peer review of "Identification of Gas-phase Pyrolysis Products in a Prescribed Fire: Seminal Detections Using Infrared Spectroscopy for Naphthalene, Methyl Nitrite, Allene, Acrolein and Acetaldehyde"

_Atmospheric Measurement Techniques, 2018_

## Referee Comment (RC1) · Anonymous Referee #1 · 11 Dec 2018

I found the paper to be both interesting and insightful. The evidence for the presence of the five gases reported on is convincing and clearly presented. Not only will this paper be useful for those interested in emissions from wildfire, but it will also be useful to those looking to model emissions from other sources utilizing infrared spectroscopy.

Two matters that could be addressed that would add to the paper's usefulness are given below.

[Figure]

-One part of the discussion that was missing is related to the comparison of the field spectra with database spectra. The PNNL database spectra are typically recorded at 5, 25 and 50 degrees Celsius whereas the spectra collected here are at 70 degrees Celsius. I would assume that the 50 C PNNL spectra would be used for comparison, but this was not mentioned. Also, do you have a metric of how the PNNL reference spectral profiles change from 5 to 25 to 50 C such that it could be said that using 50 C reference spectra to measure 70 C experimental spectra would result in an error less than 'give number'. This would also be important to mention for the pathlength calibration process since this parameter is used in the measurement of the mixing ratios of the five gases discussed.

-Also, the authors may be in a unique position to comment on whether the analysis techniques used here (i.e. MALT5) could be also used in active or passive remote infrared sensing for these gases.

---

## Referee Comment (RC2) · Anonymous Referee #2 · 21 Dec 2018

Overall this paper presents a useful extension of FTIR analysis for VOCs in samples taken from challenging environments. The paper is well written and requires few, if any, typographical corrections.

However, I do have some suggestions that I believe would increase the readability and utility of the paper.

The term "Seminal Detections" is correct but, unfortunately, the term is much more

commonly associated with seminal fluid detection and may reduce the "searchability" of the paper. Perhaps "First Detections" could be used in the title and in equivalent places within the body of the manuscript.

Lines 16/17 (first sentence in abstract): split into two sentences at the semicolon.

Line 29 "significantly": in a measurement and analysis paper the term "significantly" should be retained for use only in the statistical context. While clear enough here, it is best to avoid it and use, perhaps, "profoundly", "strongly", etc.

Line 31 "however, prescribed ...": "however" implies "in contrast to", but this sentence seems more to be about additional sources of VOCs. Perhaps "Additionally, prescribed ..."

Line 55 "thermal decomposition (pyrolysis)": the technical definition of pyrolysis seems to be something like physical and chemical transformation induced by high temperature INERT atmosphere (absence of oxygen). I am not sure that wildfire processes meet that definition. Perhaps the authors can expand upon this section slightly to make it clear that this is actually pyrolysis and not simply emission due to pressure increases (gas expansion), vapourisation, etc.

Line 85-90 Table 1: The order in which the compounds are presented in the table is not justified. A statement regarding this would be appreciated.

Line 114 "Beer's Law": should not be possessive - "a Beer Law plot". Preferable would be "a Beer-Lambert Law plot" (the Beer law is concentration only, Lambert Law is path-length only, Beer-Lambert Law incorporates both)

Line 118-122: Some further details on the calibration would be appreciated. There is no intercept reported - was the regression constrained to a zero-intercept? What is the uncertainty in the slope/pathlength?

Line 126/127: Dry-air generators are not necessarily CO2-free generators. I assume that this system does produce dry, CO2, CO and hydrocarbon-free air, but a statement

to that effect should be included.

Line 184 Figure 2: This comment applies to all figures containing stacked spectra. The colours are not easily distinguishable and colour-blind people may have some difficulty identifying the correct trace. It would be better to label each trace with a letter, (ab), (b), (c), etc, and include the legend in the figure caption.

Line 191 and following: The authors use the term "Limit of Detection" or "Detection Limit" in the sense of being the lowest concentration for which they can provide a value. However, the LoD is the limit at which the analyte can be reasonably stated as being present, but without quantification. The Limit of Quantification, the lowest concentration for which it is acceptable to provide value, is generally given as 10 x StdDev. The authors should clarify and confirm what concept they are employing.

Line 258 Figure 3 and accompanying text: The inclusion of naphthalene in the fit has not only improved the fit in the wavenumber range of the naphthalene peak (785 – 775 cm-1) but also in the 770 – 760 cm-1 range. While a minor point, it is probably worth commenting that the fitting process will result in poorer values for other species without all the contributors to the spectrum being included, as the intensities of those other components are distorted to try to compensate for the missing component. This observation holds for most of your spectra.

Line 275 – 279 Table 3: This table seems to be somewhat misplaced, being in the "3.1 Naphthalene" but referring to all compounds. It would perhaps be better placed in "3.5 Comparison to Other Sections" (the section can be renamed to something like "Limits of Detection and Comparison to Other Methods".

Line 280-287 and corresponding text for other compounds: Each section on a compound finishes with a paragraph of health aspects and atmospheric chemistry of the compound. This feels to be tacked-on and does not directly relate to the science being discussed. It would be better if these sections, if included at all, were collected and condensed to a single space, perhaps in the introduction.

Line 309 – 312: In the discussion of the conformers of methyl nitrate, it is not obvious if two spectra, one for the cis- and one for the trans-isomer, were employed. It would be appreciated if this was discussed.

Line 451 and following Section 3.5: In this section, it would be appropriate to include indicative limits of quantification

Line 464-466 Table 4. The way the units are presented in the header row (in parentheses) and the way the uncertainty of the values are presented in the table body (in parentheses) caused some confusion. IUPAC requires that parentheses are NOT used for units, rather the solidus (divide-by symbol) is used. However, I suspect that this is too radical-a-step for the journal. The uncertainty in the table would be clearer if expressed in the form "$a \pm \varepsilon$ ". Note that ppb/ppm is the same as per mille (‰.

It would be appreciated if here, or perhaps earlier, the justification for using emission ratios is given.

Section 3.5: It would be appropriate in this section to present the LoDs or LoQs of other techniques and compare this with your FTIR method. In addition, it would be appropriate to include a short comparison of sample-handling requirements, the time required, and other advantages/disadvantages of FTIR over other methods.

---

## Referee Comment (RC3) · Anonymous Referee #2 · 9 Jan 2019

Thank you. Your corrections and modifications have adequately addressed the points raised in my review.

---

## Author Comment (AC3) · 14 Jan 2019

We thank the reviewer for the constructive comments. The manuscript is being revised accordingly.

---

## Author Response (AR1)

15 January 2019

Dr. Frank Hase, Assoicate Editor *Atmospheric Measurement Techniques (AMT)* Karlsruhe Institute of Technology (KIT), Institute of Meteorology and Climate Research Hermann-von-Helmholtz-Platz 1 76344 Eggenstein-Leopoldshafen Germany

Dear Dr. Hase:

Please find enclosed the revision for manuscript amt-2018-346 titled, "Identification of Gas-phase Pyrolysis Products in a Prescribed Fire: First Detections Using Infrared Spectroscopy for Naph-thalene, Methyl Nitrite, Allene, Acrolein and Acetaldehyde." We appreciate both the referees' insighful comments, and have taken them into account in our revision. The referees' comments are addressed in turn, and we have highlighted the revised sections of the text with a red font. Please note that the "Author Comments" and "Change(s) Made" below are similar to the author comments that were posted to the interactive discussion, https://doi.org/10.5194/amt-2018-346.

We believe this study will be of interest to the atmospheric community, especially those that use infrared spectroscopy to study polluated environments, as it demonstrates the ability to detect five molecules that previously have not been observed from fire emissions using this techniuqe. We look forward to hearing from you regarding this manuscript

Sincerely,

Timothy J. Johnson, Chemist

Coauthors: Nicole K. Scharko, Post Doctorate, Ashley M. Oeck, Post Bachelors, Russell G. Tonkyn, Chemist, Steve P. Baker, Chemist Emily N. Lincoln, Chemist Joey Chong, Physical Science Technician Bonni M. Corcoran, Biological Science Technician Gloria M. Burke, Forestry Technician David R. Weise, Research Forester Tanya L. Myers, Chemist Catherine A. Banach, Post Bachelors

**Responses to Referee Comments for amt-2018-346**

**Referee #1**

**1. Referee Comments:** I found the paper to be both interesting and insightful. The evidence for the presence of the five gases reported on is convincing and clearly presented. Not only will this paper be useful for those interested in emissions from wildfire, but it will also be useful to those looking to model emissions from other sources utilizing infrared spectroscopy. Two matters that could be addressed that would add to the paper's usefulness are given below. One part of the discussion that was missing is related to the comparison of the field spectra with database spectra. The PNNL database spectra are typically recorded at 5, 25 and 50 degrees Celsius whereas the spectra collected here are at 70 degrees Celsius. I would assume that the 50 C PNNL spectra would be used for comparison, but this was not mentioned.

**2.** Author Comments: We thank the referee for the review and insightful comments. The referee is correct, the 50°C PNNL reference spectra were used for analysis, and that was not explicitly mentioned in the text.

**3. Change(s) Made:** We have revised the text to include the following, "The PNNL database provides reference spectra measured at 5, 25 and 50 °C, all of which have been normalized to a number density of 296 K (~23 °C) and 1 atmosphere. While not perfectly optimal, the PNNL 50 °C reference spectra were used for evaluation to best match the bandshapes of the 70 °C experimental data."

**1. Referee Comments:** Also, do you have a metric of how the PNNL reference spectral profiles change from 5 to 25 to 50 C such that it could be said that using 50 C reference spectra to measure 70 C experimental spectra would result in an error less than 'give number'.

2. Author Comments: This is an excellent point, and we thank the referee for the comment.

3. Change(s) Made: Section 2.3 has been revised to include the following,

"The fit of the 50 °C PNNL reference data to the 70 °C experimental spectra is obviously less than ideal. To correctly fit to the experimental spectra, reference data at 70 °C are needed, but short of this knowledge of the temperature, partition function and individual line assignments are needed, and this changes for each line or set of lines for each molecule used in the fit. While MALT5 correctly accounts for gas temperature in all cases and for intensities of the HITRAN line-by-line data, it cannot do so for the PNNL reference data. At higher temperatures, there can be increases in population/intensity of the high-J lines with decreases for the lines originating with low J values. The effect is more pronounced for smaller, more rigid molecules (e.g. allene, acetaldehyde) than for the bands associated with larger, less rigid molecules of low symmetry. Preliminary estimates for the quality of fit estimate errors in the 2 to 5 percent range, though the value depends strongly on the species and which waveband is used for the fit."

**1. Referee Comments:** This would also be important to mention for the path length calibration process since this parameter is used in the measurement of the mixing ratios of the five gases discussed

**2.** Author Comments: The path length calibration was conducted at room temperature, and we have updated the text.

**3.** Change(s) Made: We have included the following text in Section 2.2, "Measurements conducted at room temperature of pure isopropyl alcohol (IPA, Sigma-Aldrich, 99.5%) at ten different pressures were collected and a Beer-Lambert Law plot was created to determine the length."

**1. Referee Comments:** -Also, the authors may be in a unique position to comment on whether the analysis techniques used here (i.e. MALT5) could be also used in active or passive remote infrared sensing for these gases.

**2. Author Comments:** We appreciate the referee's comment, and we have updated the manuscript with the following text along with citations to demonstrate the versatility of MALT.

**3.** Change(s) Made: Section 2.3 includes the following, "The MALT analysis technique has previously been used in both open-path and extractive FTIR systems with active sources. (Burling et al., 2010; Burling et al., 2011; Akagi et al., 2013; Akagi et al., 2014). The program has also been used for ground-based solar FTIR measurements (Griffith et al., 2003)."

**References**

- Akagi, S. K., Yokelson, R. J., Burling, I. R., Meinardi, S., Simpson, I., Blake, D. R., McMeeking, G. R., Sullivan, A., Lee, T., Kreidenweis, S., Urbanski, S., Reardon, J., Griffith, D. W. T., Johnson, T. J., and Weise, D. R.: Measurements of reactive trace gases and variable O3 formation rates in some South Carolina biomass burning plumes, Atmos. Chem. Phys., 13, 1141-1165, 2013.
- Akagi, S. K., Burling, I. R., Mendoza, A., Johnson, T. J., Cameron, M., Griffith, D. W. T., Paton-Walsh, C., Weise, D. R., Reardon, J., and Yokelson, R. J.: Field measurements of trace gases emitted by prescribed fires in southeastern US pine forests using an open-path FTIR system, Atmos. Chem. Phys., 14, 199-215, 2014.
- Burling, I. R., Yokelson, R. J., Griffith, D. W. T., Johnson, T. J., Veres, P., Roberts, J. M., Warneke, C., Urbanski, S. P., Reardon, J., Weise, D. R., Hao, W. M., and de Gouw, J.: Laboratory measurements of trace gas emissions from biomass burning of fuel types from the southeastern and southwestern United States, Atmos. Chem. Phys., 10, 11115-11130, 2010.
- Burling, I. R., Yokelson, R. J., Akagi, S. K., Urbanski, S. P., Wold, C. E., Griffith, D. W. T., Johnson, T. J., Reardon, J., and Weise, D. R.: Airborne and ground-based measurements of the trace gases and particles emitted by prescribed fires in the United States, Atmos. Chem. Phys., 11, 12197-12216, 2011.
- Griffith, D. W. T., Jones, N. B., McNamara, B., Walsh, C. P., Bell, W., and Bernardo, C.: Intercomparison of NDSC ground-based solar FTIR measurements of atmospheric gases at Lauder, New Zealand, J. Atmospheric Ocean. Technol., 20, 1138-1153, 2003.

**Responses to Referee Comments for amt-2018-346**

**Referee #2**

**1. Referee Comments:** Overall this paper presents a useful extension of FTIR analysis for VOCs in samples taken from challenging environments. The paper is well written and requires few, if any, typographical corrections. However, I do have some suggestions that I believe would increase the readability and utility of the paper. The term "Seminal Detections" is correct but, unfortunately, the term is much more commonly associated with seminal fluid detection and may reduce the "searchability" of the paper. Perhaps "First Detections" could be used in the title and in equivalent places within the body of the manuscript.

**2.** Author Comments: We thank the referee for the constructive suggestions to improve the readability, utility and "searchability" of the paper.

**3.** Change(s) Made: The term "seminal" has been replaced with "first" in the title and throughout the manuscript.

**1. Referee Comments:** Lines 16/17 (first sentence in abstract): split into two sentences at the semicolon.

**2. Author Comments:** We thank the referee for the suggestion.

**3.** Change(s) Made: Lines 16/17 have been corrected as suggested.

**1. Referee Comments:** Line 29 "significantly": in a measurement and analysis paper the term "significantly" should be retained for use only in the statistical context. While clear enough here, it is best to avoid it and use, perhaps, "profoundly", "strongly", etc.

2. Author Comments: We thank the referee for the suggestion.

3. Change(s) Made: The term "significantly" has been replaced with "profoundly" in the line 29

**1. Referee Comments:** Line 31 "however, prescribed ...": "however" implies "in contrast to", but this sentence seems more to be about additional sources of VOCs. Perhaps "Additionally, prescribed..."

2. Author Comments: We thank the referee for the suggestion.

3. Change(s) Made: The term "however" has been replaced with "additionally" in the line 31.

**1. Referee Comments:** Line 55 "thermal decomposition (pyrolysis)": the technical definition of pyrolysis seems to be something like physical and chemical transformation induced by high temperature INERT atmosphere (absence of oxygen). I am not sure that wildfire processes meet that definition. Perhaps the authors can expand upon this section slightly to make it clear that this is actually pyrolysis and not simply emission due to pressure increases (gas expansion), vapourisation, etc.

**2.** Author Comments: We thank the referee for this comment, and we have added text for further clarification. We know that pyrolysis did and does occur (and not simply vaporization of hydrocarbons) because pyrolysis generates the fuel gases that combust to produce the flame, which is what is visually observed.

**3.** Change(s) Made: The manuscript now states: "Pyrolysis is the chemical transformation of material by heat in an oxygen-free or low-oxygen environment. Wildland fire consists of multiple processes: thermal decomposition (pyrolysis) of solid wildland fuels into gases, tars, and char followed by combustion (oxidation) of pyrolysis products resulting in flame gases and particulate matter in the smoke. The visible flame is sustained by fuel gases that are produced by pyrolysis (Ward et al., 1991). These two processes (pyrolysis and combustion) are complementary given that heat released from the oxidation reactions facilitates further pyrolytic reactions allowing the fire to advance."

**1. Referee Comments:** Line 85-90 Table 1: The order in which the compounds are presented in the table is not justified. A statement regarding this would be appreciated.

2. Author Comments: We thank the referee for the comment.

**3.** Change(s) Made: Table 1 has been rearranged, and the compounds are in order by category (i.e. hydrocarbon, oxygenated hydrocarbon, nitrogen-containing species etc...) and followed by number of carbons in the molecule. Also, Table 1 now has both the name along with the formula for each compound.

**1. Referee Comments:** Line 114 "Beer's Law": should not be possessive - "a Beer Law plot". Preferable would be "a Beer-Lambert Law plot" (the Beer law is concentration only, Lambert Law is path length only, Beer-Lambert Law incorporates both)

**2. Author Comments:** We thank the referee for the suggestion to use "a Beer-Lambert Law plot". [We note that Johann Lambert was a Swiss physicist and mathematician and August Beer was a German physicist for whom the law was named. The use of the possessive form "Beer's law" is in fact correct.]

**3.** Change(s) Made: The term "Beer's law" has been replaced with "a Beer-Lambert Law plot" in line 114.

**1. Referee Comments:** Line 118-122: Some further details on the calibration would be appreciated. There is no intercept reported - was the regression constrained to a zero-intercept? What is the uncertainty in the slope/pathlength?

**2.** Author Comments: We thank the referee for the suggestion.

**3.** Change(s) Made: As suggested by the referee, we have included further details on the calibration. The axes in Figure 1b have been renamed to add more detail. The y-axis is now labeled as "Integrated Area of IPA measurements", and the x-axis is now labeled as "Mixing Ratio  $\times \varepsilon$  (area of IPA reference)". Additionally, Section 2.2 includes the name of the software used for integration as well as the following text, "The y-intercept was set to zero. The slope is equal to the path length, which was determined to be  $8.10 \pm 0.1$  m (standard error of the regression)."

**1. Referee Comments:** Line 126/127: Dry-air generators are not necessarily CO2-free generators. I assume that this system does produce dry, CO2, CO and hydrocarbon-free air, but a statement to that effect should be included.

**2.** Author Comments: The air generator produced both dry and  $CO_2$ -free air, and there was no evidence of CO nor CH4 in the single-beam background spectra. Trace quantities of H2O(g) and CO2 were present in the background spectra due to ambient air getting into the system, however, the intensities were negligible in comparison to the intensities of H2O and CO2 in the experimental spectra. Furthermore, the features observed in the single beam background spectra are likely cancelled out (or partially cancelled out) when the experimental spectrum is ratioed to the background spectrum.

**3.** Change(s) Made: We have added the following to section 2.2, "Inspection of a single beam background spectrum showed no evidence of CO or  $CH_4$  contaminants and only negligible amounts of  $H_2O$  and  $CO_2$ ."

**1. Referee Comments:** Line 184 Figure 2: This comment applies to all figures containing stacked spectra. The colours are not easily distinguishable and colour-blind people may have some difficulty identifying the correct trace. It would be better to label each trace with a letter, (ab),(b), (c), etc, and include the legend in the figure caption.

**2.** Author Comments: We thank the referee for the suggestion.

**3.** Change(s) Made: As suggested, we have labelled each trace with a number (instead of a letter because Figure 2 already has letters) that corresponds to its description in the legend. All figure were amended except for Figure 6 since all of the traces are already labelled.

**1. Referee Comments:** Line 191 and following: The authors use the term "Limit of Detection" or "Detection Limit" in the sense of being the lowest concentration for which they can provide a value. However, the LoD is the limit at which the analyte can be reasonably

stated as being present, but without quantification. The Limit of Quantification, the lowest concentration for which it is acceptable to provide value, is generally given as 10 x StdDev. The authors should clarify and confirm what concept they are employing.

**2. Author Comments:** We thank the referee for the comment. After much consideration, we have decided to use the language "Detection Limit" because it is a standard term in IR spectroscopy that represents the minimum amount of an analyte that may be detected.

We acknowledge that our approach of determining this value is unique because it is obtained from  $3 \times$  the RMS of the residual instead of  $3 \times$  peak-to-peak noise of the absorbance. As stated, we are reporting signal-to-residual, not signal-to-noise detection limits.

**3.** Change(s) Made: The following text has been added for clarity, "In IR spectroscopy, detection limits often represent the minimum amount of analyte that may be detected and are reported as two to three times the signal-to-noise (Griffith et al., 2006)."

For clarification, we have renamed section 2.5 to "Signal-to-Residual Detection Limits" and we have added the word "residual" to column 5 in Table 3.

**1. Referee Comments:** Line 258 Figure 3 and accompanying text: The inclusion of naphthalene in the fit has not only improved the fit in the wavenumber range of the naphthalene peak (785 –775 cm-1) but also in the 770 – 760 cm-1 range. While a minor point, it is probably worth commenting that the fitting process will result in poorer values for other species without all the contributors to the spectrum being included, as the intensities of those other components are distorted to try to compensate for the missing component. This observation holds for most of your spectra.

**2.** Author Comments: We thank the referee for the comment, and we fully agree. The presence of naphthalene in the analysis improves the fit, which consequently improves the values for the other species. This is mentioned in the introduction, however, it is worth noting in this section as well.

**3.** Change(s) Made: The manuscript includes the following, "Including naphthalene in the analysis clearly improves the fit, which consequently improves the derived values for the other species. This observation is consistent in the spectral analyses for all target compounds discussed below."

**1. Referee Comments:** Line 275 – 279 Table 3: This table seems to be somewhat misplaced, being in the "3.1 Naphthalene" but referring to all compounds. It would perhaps be better placed in "3.5 Comparison to Other Sections" (the section can be renamed to something like "Limits of Detection and Comparison to Other Methods".

**2.** Author Comments: We agree that Table 3 is somewhat misplaced, and we have moved it to the section before 3.1 Naphthalene. Since Table 3 is referred to in sections 3.1-3.5, we felt that it was necessary to present it earlier in the manuscript (instead of placing it in section 3.5).

**3.** Change(s) Made: We have moved it to the section before 3.1 Naphthalene.

**1. Referee Comments:** Line 280-287 and corresponding text for other compounds: Each section on a compound finishes with a paragraph of health aspects and atmospheric chemistry of the compound. This feels to be tacked-on and does not directly relate to the science being discussed. It would be better if these sections, if included at all, were collected and condensed to a single space, perhaps in the introduction.

2. Author Comments: We thank the referee for the comment.

**3.** Change(s) Made: Those paragraphs and corresponding text have been removed from the manuscript.

**1. Referee Comments:** Line 309 – 312: In the discussion of the conformers of methyl nitrate, it is not obvious if two spectra, one for the cis- and one for the trans-isomer, were employed. It would be appreciated if this was discussed.

2. Author Comments: This is an excellent point.

**3.** Change(s) Made: We have added the following text to section 3.2 for clarity, "The PNNL reference spectrum for methyl nitrite was created using an equilibrium mixture of *cis* and *trans*, and the single spectrum contains feature from both conformers (Sharpe et al., 2004)."

**1. Referee Comments:** Line 451 and following Section 3.5: In this section, it would be appropriate to include indicative limits of quantification

2. Author Comments: Please see comments below regarding section 3.5.

**3.** Change(s) Made: Please see changes below regarding section 3.5.

**1. Referee Comments:** Line 464-466 Table 4. The way the units are presented in the header row (in parentheses) and the way the uncertainty of the values are presented in the table body (in parentheses) caused some confusion. IUPAC requires that parentheses are NOT used for units, rather the solidus (divide-by symbol) is used. However, I suspect that this is too radical-a-step for the journal. The uncertainty in the table would be clearer if expressed in the form "a  $\pm \varepsilon$ " ". Note that ppb/ppm is the same as per mille (‰).

**2. Author Comments:** We thank the referee for these comments. We included the parentheses around the units not because it is too radical a step, but rather for clarity. There is a solidus in the unit symbol, and we thought it would be too confusing expressing the units as "/ppb/ppm" instead

of (ppb/ppm). We have decided not to use the unit symbol for per mille for clarity and because it is often displayed in ppb/ppm [please see Table 2 in Koss et al. (2018)]

3. Change(s) Made: We have displayed the uncertainties as the form suggested in Table 4.

**1. Referee Comments:** It would be appreciated if here, or perhaps earlier, the justification for using emission ratios is given.

**2.** Author Comments: We thank the referee for the comment.

**3.** Change(s) Made: We have included the following text for justification for using emission ratios. Section 3.5 now includes, "An emission ratio is a standard metric used in fire emission measurements and is defined as the change in the mixing ratio of the target compound relative to the change in mixing ratio of the reference species, generally either carbon monoxide or carbon dioxide (Urbanski et al., 2008). Here, carbon monoxide is used as the reference species since the present study focuses on pyrolysis, and prior fire studies generally provide emission ratios relative to carbon monoxide, which makes it a convenient quantity for comparison."

**1. Referee Comments:** Section 3.5: It would be appropriate in this section to present the LoDs or LoQs of other techniques and compare this with your FTIR method. In addition, it would be appropriate to include a short comparison of sample-handling requirements, the time required, and other advantages/disadvantages of FTIR over other methods.

**2. Author Comments:** We thank the referee for these comments, and we point out that Table 1 in Koss et al. (2018) compares (in great detail) the various techniques used for biomass burning measurements. The table contains detection limits, time resolution and inlet setup as well as other information for seven instruments (Koss et al., 2018). The FTIR technique presented in that table is open-path, which has a lower detection limit (1 ppb) compared to the extractive method used in the present study (low ppm). We also note that in interest of time and space, we have only given a quick summary of comparison to other optical techniques. If one begins a comparison of the LoD or LoQ of a few other techniques, the manuscript rapidly becomes a very lengthy review article of many other techniques, including their advantages and pitfall. In the interest of brevity, we have chosen not to go down that path.

**3.** Change(s) Made: We have included the following text to acknowledge Table 1 in Koss et al. (2018) and to include the detection limits for the extractive FTIR for comparison. Section 3.5 now includes, "There are of course advantages and disadvantages for the various measurement techniques typically used in biomass burning investigations. For a detailed summary of instrumental methods (including species measured, time resolution and detection limits), the reader is referred to Table 1 found in Koss et al. (2018). The FTIR technique presented in that table is open-path (OP-FTIR), which has a lower (better) detection limit (typically on the order of 10s of ppb) as compared to the extractive method used in the present study (low ppm, see Table 3). It should be acknowledged that the target compounds and spectral analysis methods of this study are also fully applicable to both infrared laser systems and OP-FTIR systems."

**References**

- Griffith, D. W. T., and Jamie, I. M.: Fourier Transform Infrared Spectrometry in Atmospheric and Trace Gas Analysis, Encyclopedia of Analytical Chemistry: Applications, Theory and Instrumentation, 2006.
- Koss, A. R., Sekimoto, K., Gilman, J. B., Selimovic, V., Coggon, M. M., Zarzana, K. J., Yuan, B., Lerner, B. M., Brown, S. S., Jimenez, J. L., Krechmer, J., Roberts, J. M., Warneke, C., Yokelson, R. J., and de Gouw, J.: Non-methane organic gas emissions from biomass burning: identification, quantification, and emission factors from PTR-ToF during the FIREX 2016 laboratory experiment, Atmos. Chem. Phys., 18, 3299, 2018.
- Sharpe, S. W., Johnson, T. J., Sams, R. L., Chu, P. M., Rhoderick, G. C., and Johnson, P. A.: Gas-phase databases for quantitative infrared spectroscopy, Appl. Spectrosc., 58, 1452-1461, 2004.
- Urbanski, S. P., Hao, W. M., and Baker, S.: Chemical composition of wildland fire emissions, Developments in Environmental Science, 8, 79-107, 2008.
- Ward, D. E., and Hardy, C. C.: Smoke emissions from wildland fires, Environ. Int., 17, 117-134, 1991.